# Physical Activity vs. Redox Balance in the Brain: Brain Health, Aging and Diseases

**DOI:** 10.3390/antiox11010095

**Published:** 2021-12-30

**Authors:** Paweł Sutkowy, Alina Woźniak, Celestyna Mila-Kierzenkowska, Karolina Szewczyk-Golec, Roland Wesołowski, Marta Pawłowska, Jarosław Nuszkiewicz

**Affiliations:** Department of Medical Biology and Biochemistry, Ludwik Rydygier Collegium Medicum in Bydgoszcz, Nicolaus Copernicus University in Toruń, 85-092 Bydgoszcz, Poland; al1103@cm.umk.pl (A.W.); celestyna_mila@cm.umk.pl (C.M.-K.); karosz@cm.umk.pl (K.S.-G.); roland@cm.umk.pl (R.W.); marta.pawlowska@cm.umk.pl (M.P.); jnuszkiewicz@cm.umk.pl (J.N.)

**Keywords:** physical exercise, oxidant–antioxidant equilibrium, central nervous system, neurodegeneration, exerkines, cognition, memory

## Abstract

It has been proven that physical exercise improves cognitive function and memory, has an analgesic and antidepressant effect, and delays the aging of the brain and the development of diseases, including neurodegenerative disorders. There are even attempts to use physical activity in the treatment of mental diseases. The course of most diseases is strictly associated with oxidative stress, which can be prevented or alleviated with regular exercise. It has been proven that physical exercise helps to maintain the oxidant–antioxidant balance. In this review, we present the current knowledge on redox balance in the organism and the consequences of its disruption, while focusing mainly on the brain. Furthermore, we discuss the impact of physical activity on aging and brain diseases, and present current recommendations and directions for further research in this area.

## 1. Introduction

The history of learning about the structure of the human body is probably as old as the history of human life on Earth. Certainly, it was known how the brain is structured and how nerves are connected to muscles and other organs as far back as Ancient Greece. Thus, it was known that the brain regulates our moving and sensation. Moreover, the view was held that physical activity (PA) and healthy diet are key factors in promoting health, including mental health [1]. Currently, the issue of lack of PA due to a sedentary lifestyle is raised especially in the context of civilization diseases. Most recently, however, PA effects have been taken into consideration in neurodegenerative and mental disorders, as well as in the context of brain performance. Physical effort has antidepressant and analgesic properties. It delays aging. PA improves not only physical and mental health, but also cognitive function and memory [2,3]. Interestingly, during human evolution the characteristics of the musculoskeletal system associated with endurance positively correlated with brain size. In rodents, movement-dependent food acquisition positively affects both their general physiology and their brains (improvement of cognitive abilities). Similar relationships are also observed in other animals [2].

PA primarily influences the redox balance. During physical effort, oxygen consumption and production of reactive oxygen species (ROS), particularly oxygen free radicals (OFRs), increase significantly in mitochondria. Oxidation reactions are strongly intensified. Such conditions may lead to oxidative stress, resulting in oxidative damage of nucleic acids and proteins. However, polyunsaturated fatty acids (PUFAs) are most vulnerable due to the large number of double bonds and relatively easy access to them. Negative effects of physical exercise in the form of oxidative damage are short-term, and ultimately lead to the development of defense mechanisms, which, in a general sense, are based on the increase in antioxidant capacity as a result of adaptation to increased concentrations of ROS. Regular PA increases antioxidant abilities, resulting in the enhancement of the oxidant–antioxidant balance in the organism [4,5].

Oxidative stress through molecular damage may also promote inflammation. Therefore, chronic oxidative stress leads to disorders and may result in diseases. Reactive nitrogen species (RNS) are most commonly observed in elevated concentrations. Undoubtedly, the course of most diseases is strictly associated with oxidative stress [6]. It seems that neurodegenerative diseases (ND) are especially associated with oxidative imbalance, since the brain, or in a broader sense, the nervous system, is exceptionally rich in PUFAs and is characterized by very high oxygen consumption. Additionally, the incidence of ND is strongly and positively correlated with age, while aging with oxidative stress, which results, i.a., from the fact that antioxidant abilities decrease in the organism with age [7]. Therefore, the main hypothesis we make in this review is that exercise prevents brain aging and neurodegenerative diseases mainly by enhancing the redox balance in the organism. We make an attempt to explain all the positive effects of exercise on the human organism, especially on the brain through the influence of exercise on oxidation and reduction reactions. We present the current knowledge on redox balance in humans in the context of PA, with particular emphasis on the central nervous system (CNS), its health, aging and diseases. In this aspect, we also discuss the issue of antioxidant supplementation and attempt to set recommendations and indicate further areas of research.

## 2. The Oxidant–Antioxidant (Redox) Balance

The human brain, despite its relatively low weight of about 1400 g, is characterized by a high demand for oxygen (O_2_) [8]. It has been determined that the brain consumes approximately 20% of O_2_ contained in the blood [9]. Intensive aerobic metabolism is associated with the generation of significant amounts of ROS. The major site of ROS production is the mitochondrial respiratory chain due to electron leakage [10]. This is the source of OFRs: superoxide anion radical (O_2_^•−^), hydrogen peroxide (H_2_O_2_), and hydroxyl radical (^•^OH). Those free radicals lead to oxidative modifications of cells that build CNS [11]. As a result of the ROS reaction with lipids, cell membranes are damaged in a process known as lipid peroxidation [12]. An increase in the permeability and depolarization of the cell membrane prevents neurons from fulfilling their role. Free-radical reactions concerning enzymatic and structural proteins are equally dangerous for homeostasis in the nervous system [13]. ROS also affect nuclear and mitochondrial DNA, leading to mutations [14]. Oxidative modifications at the molecular level and oxidative stress are characteristic of numerous brain diseases, including ND, neuroinflammation and carcinomas [6,7]. In the nervous system, ROS play an important role in maintaining homeostasis. In the human brain, ROS are responsible not only for negative reactions related to oxidative stress, but also for important physiological processes. Despite the very short lifetime of most ROS, they form the redox signaling system necessary for the proper functioning of the brain [13,14]. It has been indicated that O_2_^•−^ and H_2_O_2_, generated in the reaction catalyzed by NADPH oxidase 2 (NOX2), regulate the growth of adult hippocampal progenitor cells through phosphatidylinositol 3-kinase/protein kinase B (PI3K/AKT) signaling [15]. Cognitive impairment was observed in mice with blocked NOX2 activity [9]. This indicates an important role of ROS in the process of learning and memory functioning. In animal model studies, beneficial effects of H_2_O_2_ on axonal pathfinding and regeneration through the modulation of the Hedgehog protein pathway were observed [16,17]. It is particularly important for the proper development of the brain and maintaining its plasticity. Another subject of research is the participation of ROS in the angiogenesis process in the brain, especially as a result of trauma and hypoxia. So far, the course of this process has not been fully understood. Researchers indicate that ROS stabilize and augment the activity or responses to signaling factors, including vascular endothelial growth factor (VEGF) [18]. In order to maintain the proper functioning of the brain, it is necessary to eliminate malfunctioning cells by apoptosis. Oxidative stress and ROS are an initiator of apoptosis [19].

Homeostasis is extremely dependent on adequate concentrations of ROS, that is, the oxidant–antioxidant balance. It can prevent negative phenomena related to oxidative stress. At the same time, it allows physiological processes to occur, mediated by free radicals. That balance is ensured by antioxidants, molecules that are capable of neutralizing ROS and reducing the negative effects of oxidative stress (Figure 1) [20]. In the brain, but also in other tissues, there are enzymes with antioxidant properties. Especially important are superoxide dismutase (SOD), catalase (CAT), glutathione peroxidase (GPX), and glutathione S-transferase (GST) [21]. SOD, CAT and GPX scavenge ROS (O_2_^•−^ and H_2_O_2_), whereas GST is an enzyme essential for the functioning of GPX. Melatonin (N-acetyl-5-methoxy tryptamine), a sleep and circadian rhythm regulator synthesized and secreted by the pineal gland located in the epithalamus, is another particularly important antioxidant in the brain [22]. Melatonin is a direct free radical scavenger and a modulator of antioxidant enzymes. It also improves the efficiency of the mitochondrial electron transport chain [23,24].

Lastly, the nervous system contains significant amounts of lipids [25]. Especially important is docosahexaenoic acid (DHA), which accounts for 10–20% of the total fatty acid composition of the brain and has been shown to have neuroprotective effects. DHA is a type of PUFAs. These fatty acids decrease the synthesis of proinflammatory lipid mediators and participate in microglial activation and myelination. PUFAs are also involved in modulation of gene expression within the thioredoxin and glutathione antioxidant systems and related pathways [7]. At the same time, however, they are extremely susceptible to oxidative damage (lipid peroxidation). This fact along with high oxygen consumption cause the brain to be particularly exposed to oxidative stress [7,23].

## 3. The Redox Balance in Brain Disorders

The brain is an organ sensitive to oxidative stress due to its oxygen-demanding metabolism, relatively weak enzymatic antioxidant defense and high levels of oxidizable substrates, as well as catalytic transition metals, including copper and iron, characteristic of some brain regions [26,27]. It should be considered that the brain consumes almost ten times more oxygen and glucose than other tissues, having a high demand for adenosine triphosphate (ATP). As a result, mitochondria, found in high amounts in neurons, significantly contribute to the generation of ROS and, consequently, to oxidative processes in the brain. Increased ROS production disrupts normal mitochondrial function because antioxidant defense is an energy-intensive process. Thus, the intensification of antioxidant defense reduces the energy resources available for normal cell function [14]. Mitochondrial redox imbalance influences the normal functioning of neurons, thus participating in neurodegeneration. Increased oxidative stress in the mitochondria of dopaminergic neurons is believed to participate in the pathogenesis of Parkinson’s disease (PD) [28,29]. Analogous pathological processes, associated with mitochondrial dysfunctions resulting from the excess production of ROS, have been found to play a detrimental role in Alzheimer’s disease (AD), characterized by neuronal degradation in brain regions that control memory and cognitive and emotional functions [26,30,31]. The age-associated loss of mitochondrial functions is supposed to affect the expression and processing of amyloid precursor protein (APP), producing amyloid beta oligomers, which accumulate as plaques in AD [31]. It has not been conclusively established whether mitochondrial dysfunction found in the course of ND is a cause or a consequence of neurodegeneration, but it is undoubtedly involved in disease progression, influencing metabolism and functioning of neurons [32].

In the antioxidant defense of the brain, special attention should be paid to glutathione (GSH, reduced form), which is the most important endogenous antioxidant in this organ [33]. The synthesis of GSH in neurons is a process dependent on the amount of extracellular cysteine available [34]. Neurons lack systems to take up cystine, easily formed extracellularly from cysteine in the autooxidation process. Therefore, the supply of cysteine for neurons is supported by reducing the action of GSH synthesized in surrounding astrocytes, which are able to take up cystine via the cystine/glutamate exchange transporter [33]. To provide sufficient levels of GSH, the pentose–phosphate pathway (PPP) is activated in astrocytes to produce the reduced form of nicotinamide adenine dinucleotide phosphate (NADPH). Significant levels of NADPH are required to reduce the oxidized form of glutathione (GSSG) in a reaction catalyzed by glutathione reductase (GR). In diabetic patients, the activity of the PPP has been found to increase as a result of elevated glucose concentrations, which indicates the importance of the astroglial PPP in the antioxidant defense of the brain [28]. These mechanisms have been found to be involved in the development of PD. In ND, including PD and AD, the excess release of neurotransmitters, such as glutamate and dopamine, can be a significant source of ROS [35]. The degeneration of dopaminergic neurons in the *substantia nigra* in the PD pathogenesis is at least partly related to the pro-oxidative action of dopamine. Dopamine, released by dopaminergic neurons, is a molecule inducing oxidative stress in the synaptic cleft. Dopamine quinone, produced from dopamine in an autooxidation reaction, is a significant source of ROS. An in vitro study reports that enhanced astroglial PPP activity resulting from exposure to dopamine might prevent dopamine-induced neuronal cell damage [36]. This mechanism is mediated by the Kelch-like ECH-associated protein I (Keap I) and nuclear factor-erythroid-2-related factor 2 (Nrf2) system. Nrf2 is a master transcription factor, which binds to the antioxidant response element (ARE) of various antioxidant and anti-inflammatory genes, including metallothioneins (MTs) [33]. MTs are low molecular weight proteins rich in cysteine residues. Similarly to GSH, MTs are known to have strong antioxidant properties. Metallothionein deficiency in the brain results in enhanced ROS generation, increased lipid peroxidation and oxidative damage to proteins and DNA, leading to neurodegeneration [37]. The involvement of decreased GSH and MT levels in dopaminergic neurodegeneration has been proven in numerous studies [28,33,36,37]. PUFAs, which are plentifully found in the brain and are vital for its normal function, are particularly vulnerable to lipid peroxidation. That process was found to be involved in the pathogenesis of ND, including PD and AD [30,38]. Decreased levels of mono- and polyunsaturated fatty acids were noticed in presymptomatic PD subjects [39]. Lipid peroxidation in neurons leads to free-radical-induced damage, including irreversible oxidation of proteins. Subsequently, the aggregations of abnormal protein fragments, including the neurofilament light chain, the amyloid-beta (Aβ) peptide, T-tau and α-synuclein, are formed, resulting in neurodegeneration. Significantly increased levels of oxidative-stress-induced protein aggregates can be detected in blood plasma by magnetic resonance imaging (MRI), and they are strongly positively correlated with regional brain atrophy and poor neuropsychiatric performance in PD patients [38]. Oxidation products of lipids, nucleic acids and proteins were detected in blood, cerebrospinal fluid (CSF), urine and saliva in AD patients. Enhanced lipid peroxidation in the brain of AD patients results in decreased levels of membrane phospholipids, including phosphatidylcholine (PC), phosphatidylethanolamine (PE) and their precursors, participating in the pathogenic mechanisms of AD [30]. An increased level of 4-hydroxy-2-trans-nonenal (HNE) was found in the brain, in Aβ plaques and in the CSF of AD patients [40,41]. HNE is a product of arachidonic acid peroxidation and can induce apoptosis in the course of neurodegeneration, due to its ability to form protein adducts [30]. HNE is a highly reactive aldehyde and reacts, i.a., with neurofilaments, changing their conformation and functions. Oxidatively modified proteins were found in CSF not only in AD patients but also in the course of amnestic mild cognitive impairment, which suggests the presence of early oxidative damage before the onset of clinical dementia [42].

The role of oxidative stress in migraine was also shown [43]. A decreased level of total antioxidant capacity and increased concentrations of lipid peroxidation markers were observed in migraine patients [44,45]. Abnormal markers of mitochondrial function were also found in more than 30% of migraine patients [44]. Hypomagnesemia, postulated in migraine, was shown to enhance glutamatergic neurotransmission, leading to increased oxidative stress [46]. It is worth mentioning that neurodevelopmental disorders, including Down’s syndrome, autism spectrum disorder, schizophrenia and epilepsy, were associated with oxidative stress-induced dysregulation of chloride ion homeostasis, which is related to γ-aminobutyric acid (GABA) signaling [47]. Interestingly, oxidative stress is also supposed to be involved in the course of bipolar disorder (BP). Single-nucleotide polymorphisms in the antioxidant genes of SOD 2 (mitochondrial SOD) and GPX 3 (plasma GPX) were associated with brain structures in young BP patients in regions relevant to the disease [48].

It was found that some metabolic disorders related to disturbed oxidant–antioxidant equilibrium are associated with an increased risk of ND. In the rat model of sucrose-induced metabolic syndrome, balance disruption was noticed both at the systemic level and in the brain [49]. The redox imbalance was accompanied by increased expression of key proteins in the amyloidogenesis pathway. Interestingly, sleep fragmentation (interrupted sleep) was also found to induce increased oxidative stress in the brain, including higher malondialdehyde (MDA; a secondary lipid peroxidation product) concentration and decreased activity of antioxidant enzymes [50]. The increased MDA level was positively correlated with the development of anxiety-linked behavior. Moreover, chronic stress (mental and other types; stressors of everyday life) was found to contribute to the augmentation of oxidative stress in the parts of the brain involved in the development of depression and dementia in AD [51]. All the aforementioned results strongly highlight the involvement of increased oxidative stress in the degenerative processes of the brain.

Increased generation of ROS and redox imbalance was also recognized as an important mediator of cerebral ischemia/reperfusion-induced injury [34]. Therefore, it could be expected that antioxidant therapies would be the best solution for treating and preventing diseases of CNS, which are so widespread nowadays. However, despite the promising results of experiments carried out at the cellular level or in animal models, findings of most clinical trials on the efficacy of antioxidant treatment in patients are less promising [14,35]. In recent years, the effect of antioxidant foods on aging and age-associated degenerative diseases of brain has been especially widely recognized by researchers [52,53,54,55,56,57,58,59]. Natural antioxidants provide neuroprotective effects through a variety of biological actions, such as interaction with transition metals, inactivation of free radicals, modulation in the activity of different enzymes, and their influence on intracellular signaling pathways and gene expression [53]. However, as mentioned, most of the beneficial effects of antioxidants on the brain were observed in in vitro studies. This is primarily due to the fact that the use of antioxidants found in the diet in clinical studies is limited by the difficulty in reaching their active concentration in the brain [60]. Most of the currently known dietary antioxidants are restricted in their ability to cross the blood–brain barrier [61]. In view of this fact, the development of drugs with a clear antioxidant profile that would easily cross the barrier and enter the brain offers much promise. For example, Fernandes et al. [62] combined caffeic and ferulic acids (hydroxycinnamic acids) with polyethylene glycol. They are natural antioxidants, types of dietary polyphenols, that have been recognized as an effective strategy for delaying or slowing the degenerative processes in CNS. However, it was also reported that they may exhibit cytotoxic properties (in vitro studies in models of human neuronal cells and brain endothelial cells). Nonetheless, their combination with polyethylene glycol was to maintain their antioxidant properties, reduce their cytotoxicity and improve their penetration through the blood–brain barrier (better lipophilic properties) [62]. Moreover, some studies imply that vitamins might be a good alternative in the prevention of age-related neurological diseases. The most widely studied dietary antioxidants are vitamin C, vitamin E and β-carotene [63]. In a randomized clinical trial performed in patients with mild to moderate AD, a decrease in the level of F2-isoprostane (a secondary lipid peroxidation product) in the cerebrospinal fluid was observed after a 16-week treatment with vitamin E, vitamin C and α-lipoic acid. However, the antioxidants under study did not influence cerebrospinal fluid biomarkers related to amyloid or tau proteins, which suggests a need for careful assessment when longer-term clinical trials are conducted [64]. Furthermore, as free radicals were found to play a role in the expansion of ischemic stroke brain lesions, antioxidants might help reduce this damage [65]. Free radical scavengers, such as edaravone and NSP-116, were reported to improve neurological deficits, and thus may attenuate ischemic and hemorrhagic brain injuries after stroke [66,67]. It was also proven that administration of melatonin as an adjunct therapy in asphyxiated newborns may ameliorate brain injury induced by oxidative stress and lead to an improvement in the survival rate [68,69].

Thus, although new methods of prevention and treatment of neurological disorders resulting from oxidative stress, including problems resulting from aging, are still emerging, physical activity seems to be of considerable importance in this area.

## 4. The Influence of Physical Exercise on the Redox Balance

As early as over 40 years ago, the first studies appeared suggesting that physical exercise can lead to the distortion of the oxidant–antioxidant balance. The fact that oxidative stress occurs after PA is also confirmed by the most recent experimental work [70,71]. The main source of ROS in the organism during physical exertion is skeletal muscles. However, the contribution of other organs and tissues, such as heart, lungs or white blood cells, cannot be excluded [72]. The earlier studies considered mitochondria to be the dominant source of ROS in skeletal muscles and the enhanced generation of ROS was explained by the increase in oxygen consumption that accompanies increased activity of mitochondria during PA [72,73]. More recent studies, however, indicate that mitochondria produce higher levels of ROS in State 4 (basal state, nonphosphorylating conditions) compared to State 3 (active state, phosphorylating conditions) [74,75], which means that ROS generation in mitochondria of skeletal muscles decreases during PA. At the present state of knowledge, therefore, it is hard to determine unequivocally the contribution of mitochondria to ROS generation during physical exertion. Other sources of ROS in muscle fibres during physical exercises are NADPH oxidase, PLA_2_-dependent processes, and reactions catalysed by xanthine oxidase [76]. Which mechanism becomes the main source of ROS during the performed physical activity is probably dependent on the type of physical effort [77,78]. It can be, among others, aerobic and anaerobic, in relation to intensity, or acute (a single bout of exercise) and chronic (exercise training, repeated bouts of exercise), in relation to frequency. Both aerobic and anaerobic exercise can be performed in a chronic as well as an acute manner [78]. In the course of physical effort with the majority of aerobic processes, mitochondria contribute to ROS generation [73]. After physical effort characterised by increased participation of anaerobic processes in exercise metabolism, enhanced ROS generation occurs mainly as a result of ischemia/reperfusion [73,79]. What becomes the source of ROS under such conditions is the reaction catalysed by xanthine oxidase [79]. Acute exercise, both aerobic and anaerobic, leads to enhanced ROS generation [80]. Chronic physical activity, in turn, may protect against oxidative stress damage [81].

Oxidative damage to macromolecules in blood or in skeletal muscles induced by PA is observed mainly when the exercise is long-term and intensive [76]. Single intense anaerobic exercise (Wingate test) leads to an increase in lipid peroxide concentration in blood plasma in men [82]. An increase in the concentration of oxidative stress markers, i.e., conjugated dienes (CD) and thiobarbituric acid reactive substances (TBARS), was also demonstrated, for example, in untrained healthy men, 40 min after recovery from single submaximal physical exertion on an exercise bicycle [83], or in the blood of sportsmen after training [84,85]. It was also proven, however, that regular physical exercises conducted within a training session can increase the antioxidant abilities of the organism and thereby alleviate ROS levels [86,87]. The impact of training on the processes of oxidoreduction can be both positive and negative, and it depends on the training load, training specificity and the basal level of training [88]. Physical training can, for instance, lead to a compensatory increase in the activity of SOD and CAT in erythrocytes, which was demonstrated in kayakers and rowers after training in alpine conditions [89]. Gomes et al. [90], in turn, observed a reduction in oxidative stress in the skeletal muscles of rats with heart failure subjected to aerobic training of moderate intensity on a treadmill. The most recent research based on meta-analysis in elderly persons also confirms that regular aerobic exercise has a positive impact on the level of oxidative stress in blood (among others, the level of MDA and lipid peroxides decreased, and the level of SOD and total antioxidant potential (TAP) increased) [91]. It has been also demonstrated that physical exercise may have advantageous effects on brain function. The positive effects of physical exertion on the brain include, among other things, the activation of the internal antioxidant enzyme system [92]. Regular moderate aerobic exercise may promote antioxidant capacity in the brain. However, aerobic exhausted exercise, anaerobic high-intensity exercise, or various forms of combining these types of effort may lead to weakening of the antioxidant barrier in the brain [93].

Initially, it was thought that ROS and RNS have only toxic activity, and their enhanced generation results in damage to cellular components [94], including neurodegeneration [95]. More recent research, however, demonstrates that they can also act as critical signalling molecules, inducing advantageous adaptive changes in response to stress. It was shown that ROS released as mediators from systemic tissues/cells during physical training can also contribute to changes in the structure and function of the brain [96]. Regular performance of physical exercises may increase capillarization and neurogenesis via neurotrophic factors, decrease oxidative damage, and enhance repair of oxidative damage in the brain [95]. Physical exertion improves the function of the endothelium by increasing the blood flow, which leads to increased shear stress, stimulating the release of nitric oxide (^•^NO; a type of RNS) [97]. In the case of neurodegeneration, a disorder in the functioning of the blood–brain barrier, which is built of endothelial cells, is frequently observed [97,98]. The released ^•^NO plays a significant role not only in the functioning of the circulatory system but also in that of the immune and nervous systems (as a neurotransmitter) [99].

## 5. The Influence of Physical Exercise on Redox Balance in Aging and Brain Diseases

The human population is aging and, consequently, age-related neurodegenerative disorders are becoming an increasingly serious public health problem. Additionally, lack of PA, overeating and sleep disturbances can promote neurodegeneration. These are agents that favor brain aging, as well as mental or neurological diseases [100,101]. As early as 65 years ago, Dr. Denham Harman showed ROS as a source of aging in his free radical theory of aging [102]. Nowadays the theory is still valid. It has only been extended to include free radical-induced mitochondrial DNA mutations [103]. The theory assumes gradual accumulation of oxidative damage to be a fundamental factor of cellular aging. Basically, senescence and degenerative diseases are attributed to the deleterious action of free radicals on cell constituents and connective tissues. In general, inside the cell, free radicals are generated in reactions catalyzed by oxidoreductive enzymes involving molecular oxygen, whereas in connective tissues by transient metals [102]. There are many studies that demonstrate ROS overproduction in the organism. Rodriguez-Manas et al. [104] postulated that age-dependent endothelial dysfunction is a complex process that involves several pro-oxidant and pro-inflammatory mediators. They found that the main source of oxidative stress in nonpathological vascular aging is NADPH-dependent superoxide production. It was revealed that the activity of NADPH oxidase and the expression of NF-κB (the major transcription factor in the regulation of the response to oxidative stress) increase with age in humans [104]. Progressive redox imbalance decreases the functionality of neurons and increases the prevalence of ND. ROS, however, are also fundamental in redox signaling as second messengers. They are strictly related to Ca^2+^-mediated signaling and other critical intracellular pathways. Proper concentrations of OFRs are a basic condition for maintaining homeostasis [105]. As far as the free radical theory of aging is concerned, two main strategies seem to be most important in achieving that goal, namely the limitation of ROS production and strengthening of the antioxidant barrier [106]. Presently, many researchers believe that antioxidants counteract the harmful effects of aging and neurodegeneration [105]. Nevertheless, as already mentioned in Section 3, clinical trials have yielded disappointing and often confusing (mutually exclusive) findings, in contrast to the expected benefits of an antioxidant-based diet [14,35]. As regular physical training stimulates the antioxidant barrier and leads to adaptation to excessive concentrations of OFRs, stimulation of natural antioxidant mechanisms seems to be fully justified and necessary [107,108].

There is growing evidence of the positive effects of PA on aging and brain diseases [109]. It is emphasized that lifestyle factors, including PA, play a great role in the pathogenesis of Alzheimer’s and Parkinson’s diseases. A meta-analysis showed that there is an inverse relationship between regular PA and the risk of developing AD [106]. There are also findings that indicate that aerobic fitness (endurance performance) is negatively correlated with loss of nervous tissue and cognitive deterioration with age [32]. Physical exercises induce adaptive ROS-dependent responses in the nervous system, such as proliferation and differentiation of neuronal stem cells [107]. Cognition improvement related to regular PA may be associated with an intensification of angiogenesis, synaptogenesis, synthesis of neurotransmitters and an increase in antioxidant capacity [106]. Bernardo et al. [110] observed positive alterations in mitochondrial oxygen consumption in both synaptosomal and non-synaptosomal brain mitochondria in AD-like rat models, where the rats were performing regular and long-term physical training (endurance running on a wheel). They observed neurobehavioral improvements in the rats. The authors suggested that endurance training increased mitochondrial biogenesis in the hippocampus through the activation of specific genes. PA can possibly prevent and reverse phenotypic impairments associated with AD. Interestingly, they also found that voluntarily performed PA (irregular and non-standardized) was not able to counteract AD-related deleterious consequences [110]. Mitochondria play a pivotal role in the mechanisms involved in cell death. They have a crucial role in the redox balance, regulate apoptotic pathways and contribute to the regulation of synaptic plasticity (a role in neurotransmission). They are also involved in the regulation of intracellular calcium concentration. Impairment of mitochondrial function results in cellular alterations ranging from subtle changes to cell death and tissue degeneration. Supra-physiological production of mitochondrial ROS due to a defective scavenging system is associated with aging and age-related diseases of the brain, as mentioned in Section 3 [111].

Exerkines are potentially the most important mediators of the neuroprotective effects of exercise [112] (Figure 2). These are the substances produced and secreted by various tissues into the peripheral blood during physical exertion, and which affect the entire organism [113]. For example, physical activity triggers the upregulation of exerkines in various tissues that directly or indirectly alleviate neurodegeneration processes in Alzheimer’s and Parkinson’s diseases [112]. Among these substances an important role is played, i.a., by compounds that affect the redox balance. The probable exercise-induced neuroprotection may result from upregulated antioxidant defense (particularly SOD) and may be related to beneficial mitochondrial adaptations, as well as to an increase in concentrations of other exerkines, including brain-derived neurotrophic factor (BDNF), insulin-like growth factor 1 (IGF-1), nerve growth factor (NGF) and VEGF, which are also dependent on redox signaling. Other exerkines that affect oxidant–antioxidant equilibrium include: irisin, adiponectin (ADN), fibronectin type III domain containing 5 (FNDC5), neprilysin (NEP) and insulin-degrading enzyme (IDE) [112]. Irisin is the best studied of the aforementioned. This is a skeletal muscle-secreted myokine, produced in response to physical exercise, which has protective functions in both the central and the peripheral nervous systems, including the regulation of BDNF [114]. Irisin enters CNS through the blood–brain barrier, and enhances BDNF synthesis and release, leading to augmented neuroplasticity achieved by the collaboration of irisin and BDNF [114,115]. Moreover, the protein is expressed not only in skeletal muscles but also in the brain [116]. It has been proven, for example, that it largely inhibits brain infarct volume and reduces neuroinflammation and post-ischemic oxidative stress [114,115]. In general, the dynamic changes in exerkine levels could be used as laboratory biomarkers for monitoring the effectiveness and appropriateness of the clinically prescribed exercise interventions, thus enabling the development of customized exercise therapy for individuals of varied ages, genders, and health states [112].

Neuronal benefits resulting from PA are also mediated by ^•^NO, another neurotransmitter. The increase in ^•^NO results in an improvement of the cerebral blood flow, which is associated with increased activity of endothelial nitric oxide synthase (eNOS). Upregulated antioxidant defense in CNS, as a result of regular physical exercise, leads to a decrease in the concentration of oxidative damage markers that are neurotoxic *per se*, e.g., 8-oxoguanine (8-oxoG; a marker of DNA oxidative damage) [111] (Figure 2). The neuroprotective impact of physical training can also be assumed in PD patients in relation to aldehyde dehydrogenase (ALDH). The metabolism of dopamine results in the formation of toxic 3,4-dihydroxyphenylacetaldehyde (DOPAL) and H_2_O_2_, which are found in elevated concentrations in PD patients due to decreased vesicular uptake of cytosolic dopamine and decreased DOPAL detoxification by ALDH. Hence, it was reported that regular physical exercise increased the gene expression of ALDH in the brains of senescent female mice [107]. Moreover, Sellami et al. [117] stated that regular PA plays an important role in telomere maintenance and DNA methylation, possibly through its ability to alleviate oxidative stress and inflammation.

As already mentioned, neurodegeneration affects elderly people and PA can also alter antioxidant status in these individuals. Rousseau et al. [108] revealed increased activity of GPX in subjects aged 68.1 ± 3.1 years who were performing a minimum of three training sessions per week, with a session duration of less than 1 h each. PA of moderate intensity proved to be sufficient to improve antioxidant defense. Studied SOD and GR activities did not change in a statistically significant way [108]. Interestingly, Pérez et al. [118] reported that overexpression of antioxidant enzymes in mice did not extend their life span. Melo et al. [119], in turn, found that regular exercise bouts implemented at the early stage of neurodegeneration alleviate oxidative stress (measured by H_2_O_2_ levels and SOD activity) and improve neuronal functionality in the motor cortex in rats, which resulted from restored proteostasis. Almeida et al. [120] put forward a similar conclusion. The findings of their study confirm that PA prevents H_2_O_2_ production in rats during early neurodegeneration. However, the mechanism still remains unclear.

To date, despite a large number of studies on the effects of PA on the oxidant–antioxidant system, fully consistent conclusions have not been drawn. It is evident that physical exercise can also negatively affect the human organism, including the aging process, especially in aging individuals [121], because of lower antioxidant capacity, as mentioned [7,121]. Basically, high-intensity and prolonged exercise leads to redox imbalance, which has been shown to cause disorders in skeletal muscles and peripheral fatigue [121] (Figure 2). The evaluation of optimal PA intensity seems to be especially important, as it should be appropriate for the individuals who are the focus of the research.

## 6. Conclusions

It has long been known that PA together with healthy diet are key lifestyle factors that promote health, including brain health [1]. PA delays aging and improves cognition and memory [2,3]. However, exercise bouts may generate large amounts of ROS and RNS, and free radicals are considered to be the main source of molecular damage to cellular constituents resulting in aging [76]. There is no certainty as to whether free radicals are or may be also ethological agents. Nevertheless, most diseases and many disorders are associated with oxidative stress, the most common disturbance of the oxidant–antioxidant balance [6]. The antioxidant defense is inversely dependent on aging—the more advanced age, the weaker the defense. ND, in contrast, are directly dependent on age [7]. Interestingly, PA as a potential source of oxidative stress can lead to beneficial adaptation mechanisms, which are based on increased antioxidant capacity [4,5]. In general, physical exertion also positively affects the redox balance in elderly persons [108]. It was found that ROS released from systemic tissues during PA can contribute as mediators to changes in the structure and function of the brain [96]. Regular performance of physical exercises may, via neurotrophic factors, increase capillarization and neurogenesis, as well as decrease oxidative stress and enhance repair of oxidative damage in the brain [95]. Moreover, regular exercise bouts improve blood supply to the brain (increased ^•^NO concentration) [97] (Figure 2). Lastly, the musculoskeletal system is positively correlated with brain size [2], and endurance performance is negatively correlated with loss of brain tissue and cognitive deterioration with age in humans [32]. This can result from the need of our remote ancestors to move in order to acquire food. This assumption is supported by research in rodents and other animals [2]. All of the aforementioned findings suggest that regular physical training can be a powerful tool for the maintenance of proper brain function throughout the lifespan. This also applies to brain disorders, principally to their prevention or delaying/alleviating their symptoms. Unfortunately, it is extremely difficult to implement regular PA in the case of patients suffering from mental or neurodegenerative diseases, due to their restricted mobility in general. For this reason, there are very few direct studies on the impact of PA on these individuals. Effects of PA on the oxidant–antioxidant system in humans have been mainly studied in young and middle-aged adults. Therefore, conclusions on this issue are not fully applicable in the context of patients with neurodegenerative diseases. Apart from that, physical exercise can negatively affect the human organism as far as aging processes are concerned [121]. Thus, the evaluation of optimal PA intensity and appropriate selection of exercise type seem to be of special importance, and should be set individually.

The issue of whether antioxidant supplementation should be advised remains in question. According to current knowledge, it may seem justified and necessary [52,53,54,55,56,57,58,59]. However, there is a problem with the permeability of antioxidants through the blood–brain barrier. It is insufficient to obtain a significant antioxidant effect on neurons. In addition, most studies have been performed in vitro, and clinical trials have shown inconsistent outcomes [14,35]. Nonetheless, promising results have been provided by research on the chemical modification of natural compounds known for their high antioxidant abilities [62]. Moreover, there is debate as to whether consuming large amounts of antioxidants in supplement form actually benefits health [122]. Antioxidants themselves may act as pro-oxidants in some cases and aggravate pathological processes [123]. In 2013, renowned Nobel laureate James Watson warned that antioxidants in late-stage cancers can promote cancer progression [124]. Moreover, antioxidant supplementation may suppress the synthesis and formation of endogenous antioxidants and other cell adaptation mechanisms, such as better energetic metabolism [125]. Mentor and Fisher [126] demonstrated that an excessive intake of antioxidants disturbs blood–brain barrier functionality and angiogenic properties, as well as impairs the repair function of brain capillaries, compromising the patient’s recovery. Many plants and fruits contain potent natural antioxidant compounds that can protect cells against oxidative stress [54]. At the same time, PA has been proven to be effective in enhancing the oxidant–antioxidant balance [4]. Therefore, it seems that the best solution is to use a diet rich in natural antioxidants along with regular, albeit moderate, PA.

## Figures and Tables

**Figure 1 antioxidants-11-00095-f001:**
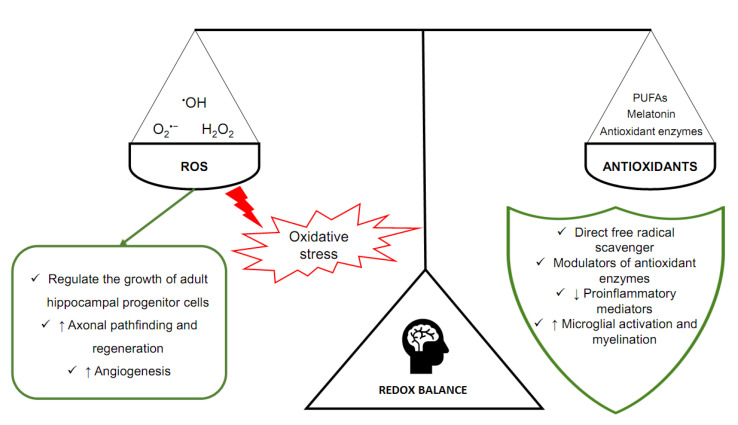
The main components of oxidoreduction equilibrium in the human brain.

**Figure 2 antioxidants-11-00095-f002:**
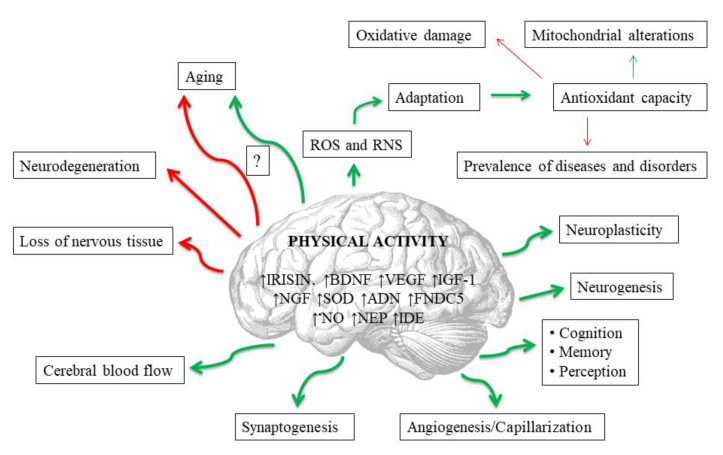
Potential impacts of regular physical activity on brain in human. Red lines mean negative impact, and green ones positive. Abbreviations are listed at the end of the article.

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
