# Peer review of "Physical Activity vs. Redox Balance in the Brain: Brain Health, Aging and Diseases"

_antioxidants, 2021, doi:10.3390/antiox11010095_

Round 1
Reviewer 1 Report
The paper by Sutkowy et al. provides a review on the role of physical activity in the maintenance of redox balance in the brain. While the authors provide a thorough description of the various modifiers of redox balance and the different redox effects of PA in the brain, there are essential discussion topics (more current) that are not included in this paper.
Introduction: lines 50-58 I feel something is missing here; what is the connection to PA. It lacks connection with the previous paragraph and the following one. This is not a good transition.
For the section on the role of PA on redox balance: the authors here need to comment and make distinctions between acute response to exercise and chronic exercise-induced adaptations. More discussion on aerobic versus anaerobic exercise would enrich this review significantly.
For section 5: This section lacks an essential aspect of the connection between PA and brain: what about skeletal muscle/brain crosstalk? The discussion here just focused on redox balanced and in the brain, but what about the muscle-brain axis, the role of exerkines, and brain health which are released in response to exercise and has shown a role in oxidative stress (i.e. irisin and others). This is a relevant and current topic that merits discussion. We need to move away from a brain-centric discussion when discussing aging and neurodegeneration. Processes occurring outside the brain may also impact the brain (i.e., gut-brain axis).
For section 6: I am not sure this discussion is pertinent here. What is the connection to PA? Overall, this is not novel/relevant information. Since PA is the focus of this manuscript, a discussion of exerkines seems more appropriate. Even the antioxidant recommendations/discussion is not novel information. Quercetin is recognized as a senolytic. This would be a more current topic to cover in this review. Along with other drugs promoting healthy aging – such as senescence-associated secretory phenotype (SASP) inhibitors and nutrient signaling regulators- and targeting redox balance.
Reviewer 2 Report
I have reviewed with a great pleasure this manuscript which aimed to discuss the interaction of physical activity (PA) with redox balance in health and disease, while focusing mainly on the effect on brain. It is a well written manuscript which provides an extensive review on this subject. However, there are some shortcomings that should be addressed by the authors:
- Section 5. The influence of physical exercise on redox balance in aging and brain diseases, page 8, lines 378-379. The authors briefly mention that “physical exercise can also negatively affect the human organism, including the aging process”. Although it was discussed in the previous section, this statement should be further detailed as it can be misleading.
- In the same Section 5, page 8, line 375 the authors state that “Effects of PA on cognition in humans have been mainly studied in young and middle-aged adults”. This statement is repeated in the Conclusions section. This is not entirely correct. Studies on the effect of PA on cognition are also available in elderly and in patients with AD and PD. These studies should be briefly discussed and not rapidly dismissed as not investigated.
- Section 6. Antioxidant supplementation. I do not understand the presence of a standalone section on antioxidants supplementation as the purpose of this review is to discuss the interaction of PA with redox balance. I suggest to briefly include some of these statements in other sections if the authors feel that they are important, although this is not the subject of the manuscript.
